# Self-regulated learning strategies and academic achievement in South Korean 6th-graders: A two-level hierarchical linear modeling analysis

**Cheyeon Ha** [1]*, **Alysia D. Roehrig**[2], **Qian Zhang**[2]

**1** The Child Study Center, Yale School of Medicine, Yale University, New Haven, Connecticut, United States of America, **2** Department of Educational Psychology and Learning Systems, Florida State University, Tallahassee, Florida, United States of America

* cheyeon.ha@yale.edu

## Abstract

The aim of the study was to explain the relationships between self-regulated learning strategy use and academic achievement of 6th-grade students in South Korea. An existing database (i.e., the Korean Educational Longitudinal Study; KELS) with 6th-grade students ($n = 7,065$) from 446 schools was used to run a series of 2-level hierarchical linear models (HLM). This large dataset enabled us to consider how the relationship between learners' self-regulated learning strategy use and academic achievement may differ at individual and school levels. We found that students' metacognition and effort regulation positively predicted their literacy and math achievement both within and across schools. The average literacy and math achievement were significantly higher in private schools than in public schools. Also, the math achievement of urban schools was significantly higher than in non-urban schools when controlling other cognitive and behavioral learning strategies. This study on 6th-grade learners' self-regulated learning (SRL) on academic achievement explores how their SRL strategies may be different from the features of successful adult learners from the previous findings, offering new insights into the development of SRL in elementary education.

## Introduction

Students who can effectively regulate their emotions, thoughts, and behaviors in their learning environments tend to have more advantages in their academic achievement [1–3]. For those reasons, self-regulation skills have been considered an important key to explaining students' learning success [4]. In the initial research on SRL, scholars found that self-regulated learners tend to use effective cognitive strategies (e.g., rehearsal, elaboration, organization, and metacognition) and behavioral strategies (e.g., regulation of task, effort, time, environment, and help-seeking) in their learning [5–7]. Pintrich's model [8, 9] of self-regulated learning, widely used in SRL studies, comprises both learning strategies and motivational beliefs. For this

Longitudinal Study (KELS) and were authorized to use data by the KEDI. Researchers should submit a research proposal to get approval for the data use from the KEDI. The institute can provide de-identified data after reviewing the research proposal. The data requests can be sent to this contact information: rsdb@kedi.re.kr.

**Funding:** The authors received no specific funding for this work.

**Competing interests:** We have no known conflict of interest to disclose.

study, we used secondary data from Korean Longitudinal Educational Study (KELS) database, which includes learning strategies basis on Pintrich's SRL model.

Despite the importance of self-regulation skills in learning, the K-12 curriculum does not often encourage teaching or practicing these skills [10]. Moreover, most previous studies of learners' self-regulation of academic achievement have been carried out in the context of higher education. Some evidence-based studies investigated children's self-regulation abilities at elementary and middle school levels (e.g., [6, 11–14]); however, small sample sizes present potential limitations to the generalizability of the prior findings. Findings of the literature highlight still a lack of evidence of the early development of elementary education and the studies conducted in the Asian context. Hence, empirical research that provides insights into the development of SRL strategies for young learners through large-scale data analysis would be highly valued.

There are various views on the early development of children's SRL strategies. Dignath, Buettner [15] pointed out that SRL strategies can be taught as early as elementary school when students are at an emotional and cognitively fluid state of development. Empirical studies on the development of SRL strategies of children aged 10 through 13 indicate that students' self-regulation skills can be a meaningful factor in explaining their academic performance [16–18]. For example, compared to 3rd-grade students, 6th-grade students are better able to use advanced SRL strategies that enhance their learning [19]. These findings support that elementary students can learn SRL strategies in schools from various learning experiences.

For this study, we used national data from the KELS, a nationally representative dataset focusing on K-12 students' learning outcomes and relevant psychological factors. To understand recent educational issues in K-12 settings, the KELS collected data on students' demographic information, school environments, academic achievement, and other educational factors. To date, previous studies using the KELS database have mainly focused on students' outcomes in secondary education. For example, Shin, Seo [20] reported that parents' and teachers' expectations were significantly related to grade 7–9 students' learning growth. You, Dang [21] found that middle students' intrinsic motivation moderated the relationship between teachers' beliefs and students' learning growth rates. Additionally, Lim and Lee [22] reported that students' career maturity was a meaningful predictor of self-management skills in elementary school, while high school students' academic achievement significantly predicted their self-management skills. In this study, we focused on 6th-grade elementary students which were not explored in the previous study and investigated the relationship between their SRL strategy use and academic achievement at both individual and school levels.

## Background

### Learning strategies of self-regulated learners

Many scholars have theorized that individual differences in self-regulation are key to explaining the learning outcomes of successful learners [23–25]. In contemporary education, using SRL skills to explore information and apply knowledge is becoming increasingly important [26]. Self-regulated learners show higher academic performance because they can analyze tasks strategically, set appropriate goals, and manage their thoughts and strategies effectively [27, 28]. Based on Pintrich [8, 9]'s explanations of self-regulated learning model, we defined the features of successful self-regulated learning as more frequent use of two types of learning strategies: 1) cognitive strategies and 2) behavioral strategies.

**Cognitive Strategies (CS) in learning.** Scholars noted that learners may use specific cognitive strategies to build up long-term memory. Moreover, metacognition involves coordinating other cognitive strategies, as a high-level cognitive skill, and it helps students effectively use

cognitive strategies in the process of evaluating their learning [29]. Bransford [30] pointed out that awareness of strengths and limitations in cognitive strategies helps the student learn and cope with learning resources flexibly. Based on Pintrich [8], Pintrich, Smith [9]'s SRL model, we explained learners' cognitive learning strategies according to four specific types: 1) rehearsal, 2) elaboration, 3) organization, and 4) metacognition.

First, rehearsal strategies are basic-level cognitive skills that load information between working memory and long-term memory. Repetitive exposure involves cognitive processes that help learners recognize important information in working memory and form links in long-term memory [31]. Second, elaboration strategies bridge new information and old knowledge, through processes like paraphrasing and summarizing. Elaboration strategies are processes of transforming information to make it more meaningful and memorable (e.g., application and analysis); thus, elaboration requires more complex cognitive processes compared to rehearsal [31]. Third, organization strategies make meaningful chunks of complex information. Self-regulated learners sense how to reduce complexity and make information well-organized and meaningful. Lastly, metacognitive strategies are higher-level cognitive skills, often defined as thinking about thinking or knowing about knowing [5]. Students use metacognition for planning, monitoring, and evaluating learning processes.

**Behavioral Strategies (BS) in learning.** Successful self-regulated learners tend to use educational resources adequately and to use higher-level of behavioral regulation skills while they are learning [28]. According to Pintrich, Smith [9] SRL model, learners with high levels of self-regulation show more strategic skills in time-, effort-, and environmental regulation. Additionally, the model includes help-seeking behaviors, which can enhance collaboration and cooperation and help students better problem-solving in the classroom. Similarly, other scholars theorized that help-seeking skills contribute to increasing productive communications and meaningful learning opportunities that allow students to access scaffolding [32, 33]. In this study, five types of behavioral learning strategies were assessed: task-regulation, effort-regulation, time-regulation, environment-regulation, and help-seeking.

First, self-regulated learners use task regulation skills to check their learning progress by preparing, planning, and evaluating task processes [8]. Second, effort-regulation is a learning skill that helps to maintain endurance even in challenging learning experiences [34]. Learners with high effort-regulation skills can reflect on where effort is most needed. Third, because time is a finite resource, time-regulation ability nurtures learning effectiveness within the planned time [35]. Fourth, self-regulated learners have flexibility in external environmental factors; thus, they are less affected by external conditions and can even improve learning environments [36]. Finally, a help-seeking approach helps students overcome academic challenges, giving them access to scaffolding from peers and teachers that can enhance their learning [37].

## Self-regulated learning strategies and academic achievement

Previous empirical studies report that students' SRL strategies are related to higher academic achievement in college and high school students [38–40]. For example, Glogger, Schwonke [38] found that cognitive learning strategies (i.e., rehearsal, organization, elaboration, and metacognitive skills) were meaningful predictors to explain high school students' journal writing quality. Thiede, Anderson [39] reported that supporting metacognition contributed to college students' higher reading comprehension levels compared to the control group. Also, multiple studies found that use of SRL strategies were a significant moderator or mediator of academic achievement. For example, a study by Mega, Ronconi [41] reported that undergraduate students' SRL use had a positive relationship with their academic achievement, and it mediated a relationship between students' positive emotions and higher academic

achievement. Villavicencio and Bernardo [42] found that first-year university students' self-regulation was not a meaningful predictor of their final grade; however, the interaction effect between SRL strategy use and enjoyment (e.g., positive emotion) was a significant predictor of their final grade. For example, in a group of students with higher enjoyment, when their SRL strategy use increased, they tended to have higher final grades.

Empirical studies of early childhood and elementary students account for a small percentage of all SRL studies. A study by Kitsantas, Steen [18] examined 5th-grade students' SRL strategy use, they found that the combined scores of cognitive and behavioral strategies were a significant predictor of students' achievement in science, language art, and math. In addition, Newman and Schwager [19] found that 6th-grade students who frequently used help-seeking strategies tended to show higher math achievement. Kindergarten children who effectively regulated self-behavior and exhibited positive classroom behaviors showed higher literacy achievement [43]. Evidence also suggests that the relationship between self-regulation and achievement persists over time [44]; for example, children's behavioral regulation skills at age 7 significantly predictor their academic achievement at age 11.

Previous meta-analysis studies provide insight into the effect of self-regulation skills on students' academic performance [12, 15]. Dignath, Buettner [15] synthesized forty-eight experimental studies of SRL skill training conducted in Western cultures. They reported that after elementary students participated in SRL training programs, their academic improvement in reading/writing and math was significantly greater than in a control group. This study focused on estimating the overall intervention effect; thus, the results are limited in understanding the relationship between academic achievement and the use of specific learning strategies. Another meta-analysis study by Dent and Koenka [12] synthesized correlational results between cognitive strategy use and students' academic achievement in elementary and secondary schools. In this meta-analysis, studies with elementary students accounted for a small proportion ($k = 10$, 16%), and a majority of included studies investigated middle school students ($k = 53$, 84%). Also, the positive relationship between SRL strategies and academic achievement varied depending on subfactors. For example, students' cognitive strategy, rehearsal, organization, elaboration, and task strategy scores were positively related to their academic achievement; however, other subfactors, students' connecting and reviewing strategy use, were not linked to their academic achievement. Therefore, further evidence is needed to understand elementary students' SRL strategy use and its benefits on academic achievement in schools.

## Study purpose and research questions

This study explored South Korean 6th-graders' SRL strategy use and their literacy and its effect on math achievement. Given that students of this age are in the early stages of developing SRL strategies, it was assumed that their SRL strategy use and academic outcomes might differ from those of adult learners. We focused on the learning strategies of self-regulated learners based on Pintrich's model [8, 9], which is interpreted in a narrow range of cognitive and behavioral strategies commonly observed in students with high self-regulation. The present study answered the following research questions:

Q1. Does 6th-grade students' cognitive learning strategy use predict their academic achievement in literacy and math?

Q2. Does 6th-grade students' behavioral learning strategy use to predict their academic achievement in literacy and math?

Q3. Does school type or school location moderate the relationship between 6th-grade students' cognitive strategy use and academic achievement in literacy and math?

Q4. Does school type or school location moderate the relationship between 6th-grade students' behavioral strategy use and academic achievement in literacy and math?

## Method

### Data sources and study design

The Korean Educational Development Institute (KEDI) conducted the national educational data collection project KELS from 2013 through 2018 with a cohort group to track the students' learning growth. Using this national dataset, this secondary data analysis study was designed as a cross-sectional study. The main dataset of this study was collected in 2014 with a cohort group of 6th-graders, which includes data on students' SRL strategies. We ran a series of 2-level HLM analyses to test predictors and moderators of students' academic achievement at the individual (i.e., level-1) and school environments (i.e., level-2).

### Participants

The original KELS dataset was obtained from a total of 7,324 6th-grade students from 458 elementary schools in South Korea. We focused on the 6th-graders' SRL strategy use and academic achievement data from this dataset. Due to incomplete outcome variables or severely low response rates in some schools during large-scale data collection, we removed problematic cases to maintain data quality and completeness. For example, we excluded the cases who did not complete the survey or achievement tests; also, there were missing cases related to data collection procedures because the data was collected during multiple weeks. The excluded data proportion was less than 3.6% ($n = 259$) of the original sample ($n = 7324$), and the final dataset of this study included 7,065 students from 446 schools for the 2-level HLM analyses (Table 1).

The dataset included similar proportions of gender groups, with 3579 females (50.7%) and 3486 males (49.3%). In the study, gender was set as a dummy variable to compare females' literacy and math achievement (i.e., reference group) against males. The dataset included a total of 446 school data from urban ($k = 181$, 40.6%), suburban ($k = 161$, 36.1%), and rural ($k = 104$, 23.3%) areas. Among the schools, 434 ($n = 6717$) were public, and 12 ($n = 348$) were private. For example, the proportion of public schools is more than 95% in South Korea; as national data collection, these school types and location proportions reflect the general composition of elementary schools in Korean educational contexts. Although the number of different school environments was unequal, when groups to be compared have equal or similar variances in the outcome variable, unequal group sizes are not a concern when there is a large enough sample size in both groups [45].

**Table 1. Demographic information of participants.**

| Variables | Categories | Sample size | Proportion (%) |
|---|---|---|---|
| Gender ($n = 7065$) | Female | 3579 | 50.7 |
| | Male | 3486 | 49.3 |
| School types ($k = 446$) | Public | 434 | 97.3 |
| | Private | 12 | 2.7 |
| School locations ($k = 446$) | Urban | 181 | 40.6 |
| | Suburban | 161 | 36.1 |
| | Rural | 104 | 23.3 |

## Measures

Students' cognitive and behavioral learning strategy use was measured by self-reported surveys based on the learning strategy questionnaires by Pintrich, Smith [9]. A research team modified the terms and structures from the original questionnaire for adult learners to make it suitable for elementary students. In this study, students responded on a 4-point Likert scale (e.g., 1 = *strongly disagree*, 2 = *disagree*, 3 = *agree*, and 4 = *strongly agree*), self-report SRL surveys. Each subscale showed stable reliability levels in this study. Table 2 shows the detailed descriptive results of the survey, including mean, standard deviation, and range information. Before the data collection, a research team conducted the pilot study with a large sample for both SRL surveys and achievement measurements, and they reviewed validity [46].

**Cognitive learning strategies.** The final dataset included four cognitive learning strategy subscales with a total of 12-items. The survey assessed *rehearsal* (e.g., "I try to memorize as many things as I can when I study"; 3 items, Cronbach $\alpha$ = .78), *elaboration* (e.g., "I try to relate ideas in this subject to those in other subjects whenever possible"; 3 items, Cronbach $\alpha$ = .81), *organization* (e.g., "When I study, I try to outline the material to help me organize my thoughts"; 3 items, Cronbach $\alpha$ = .82), and *metacognition* (e.g., "When studying for this course I try to determine which concepts I don't understand well."; 3 items, Cronbach $\alpha$ = .86).

**Behavioral learning strategies.** The final dataset included five subscales (17 items total) of behavioral learning strategies: *task-regulation* (e.g., "I find or create my own learning subject or questions"; 5 items, Cronbach $\alpha$ = .89), *effort-regulation* (e.g., "Even when subject materials are difficult, I manage to keep working until I finish"; 3 items, Cronbach $\alpha$ = .88), *time-regulation* (e.g., "I make good use of my study time for subject learning"; 3 items, Cronbach $\alpha$ = .83), *environmental-regulation* (e.g., "I usually study in a place where I can concentrate on my work"; 2 items, Cronbach $\alpha$ = .68), and *help-seeking* (e.g., "I ask the teacher to clarify concepts I don't understand well"; 7 items, Cronbach $\alpha$ = .84).

**Academic achievement in literacy and math.** The KELS data evaluated 6th-grade students' *literacy* and *math achievements* through 25 questions each. The original achievement scores between 0 and 25 were transformed into adjusted scores based on the vertical scaling approach, because it has benefits on the adaptive tests, which measured different questionnaires but focusing on learning growths. The KELS data provide information about students' academic achievement growth using weights in common questions; thus, the adjusted scores could consider the individual growths as well as test effects. The range of the adjusted *literacy achievement* scores was between 94 and 281 (*M* = 201.49, *SD* = 34.88), and the range of the adjusted *math* scores was between 100 and 267 (*M* = 203.43, *SD* = 39.00).

**Table 2. Descriptive statistics (n = 7,065).**

| Variables | Subscales | *Min.* | *Max.* | *M* | *SD*s |
|---|---|---|---|---|---|
| Cognitive | Rehearsal | 1.00 | 4.00 | 2.96 | 0.57 |
| Learning- | Elaboration | 1.00 | 4.00 | 2.91 | 0.63 |
| Strategy Use | Organization | 1.00 | 4.00 | 2.92 | 0.66 |
| | Metacognition | 1.00 | 4.00 | 2.93 | 0.66 |
| Behavioral | Task-regulation | 1.00 | 4.00 | 2.93 | 0.62 |
| Learning- | Effort-regulation | 1.00 | 4.00 | 3.20 | 0.58 |
| Strategy Use | Time-regulation | 1.00 | 4.00 | 2.82 | 0.64 |
| | Environmental-regulation | 1.00 | 4.00 | 3.02 | 0.62 |
| | Help-seeking | 1.00 | 4.00 | 2.94 | 0.54 |
| Achievement | Literacy | 94.00 | 281.00 | 201.49 | 34.89 |
| | Math | 100.00 | 267.00 | 203.43 | 39.00 |

## Data analysis

We ran a series of 2-level Hierarchical Linear Modeling (HLM) models to address our research questions, considering both the student-level (level-1) and school-level (level-2) predictors. We investigated four different HLM models based on each of the learning strategies of successful self-regulated learners (i.e., cognitive or behavioral learning strategies) and academic achievement outcomes (i.e., literacy or math).

In level-1, the following sub-factors were included as predictors in the final models: gender, and either cognitive strategy use (rehearsal, elaboration, organization, and metacognition) or behavioral strategy use (task-, effort-, time-, environment-regulation, and help-seeking). We decided to include gender as a covariate because it may be related to the different linguistic development of school-age children. All the level-1 predictors except *gender* were group-mean centered so that the random effects of these predictors on literacy and math achievement are within-school [47]. The reference gender group was set as *female*.

In level-2, we tested the moderation effect of the school environments, including school type (i.e., public or private) and location (i.e., urban, suburban, and rural), on the relationships between SRL strategy use and academic achievement in our four initial models. Specifically, we kept variables that had significantly different effects across schools, then added the school environmental variables at level-2 to explain the differences in these effects. For school type, *private* school was set as the reference group. For school location, we set *rural* as the reference group and had two dummies for *urban* and *suburban*. We describe how we build the initial and final HLM models in the result section and only report the final HLM model results in the tables.

## Results

### Descriptive statistics

Table 2 shows descriptive statistics of students' SRL features and academic achievement. For cognitive learning strategies, the averages of the four subscales were similar, about 2.9 on a 4-point Likert scale. In behavioral learning strategies, the means of the subscales were between 2.93 and 3.20, and *effort-regulation* ranked higher than others. The distribution of *math* achievement of 6th-graders showed a greater standard deviation compared to *literacy*. This means that students' individual differences were greater in *math* compared to *literacy*.

### HLM analysis results

**Unconditional models.**   First, we obtained the proportion of between-school variability of academic achievement in literacy and math. We set the unconditional model with the outcome variable of academic achievement as below.

*Level 1*: $(LITERACY; MATH)_{ij} = \beta_{0j} + r_{ij}$

*Level 2*: $\beta_{0j} = \gamma_{00} + u_{0j}$

Regarding literacy achievement, the Intraclass Correlation Coefficient (ICC) was 0.143; thus, about 14% of the variability in literacy achievement was between schools. In addition, regarding math achievement, the ICC was .195 or about 19% of the variability in math achievement between schools. According to the ICC results, we concluded that the multilevel data structure should be accounted for using HLM models.

**CS on literacy model.**   The results of the initial HLM analysis with the cognitive learning strategies as the only predictors showed that metacognition positively predicted students' literacy achievement both within and between schools. This suggests that for students in an

average school, as *metacognition* increased, their literacy achievement tended to increase, holding constant other predictors ($\gamma = 5.657$, $p < .001$); also, schools with higher average levels of *metacognition* were associated with higher average levels of *literacy* achievement ($\gamma = 13.470$, $p < .01$). *Elaboration* and *organization* significantly explained the within-school difference, yet not a between-school difference in literacy achievement, holding constant other predictors. More specifically, on average across schools, as students' *elaboration* increased, their literacy achievement also tended to increase ($\gamma = 5.079$, $p < .001$); in contrast, as students' *organization* increased, their *literacy* achievement tended to decrease ($\gamma = -2.767$, $p < .001$), holding constant other predictors. *Rehearsal* did not have a significant between-school effect on *literacy* achievement and its average within-school effect was also non-significant.

Based on the final model, the effects of *gender* and *organization* significantly varied across schools. Thus, we added school-level predictors in the final *CS on Literacy Model* (Table 3).

**Table 3. HLM results of cognitive strategy use and academic achievement.**

| Fixed Effect | Cognitive Strategy on Literacy Final Model | | | Cognitive Strategy on Math Final Model | | |
|---|---|---|---|---|---|---|
| | *B* | *SDs* | *p* | *B* | *SDs* | *p* |
| For INTRCPT1, $\beta_0$ | | | | | | |
| INTRCPT2 | 190.226 | 13.728 | <0.001*** | 190.962 | 16.725 | <0.001*** |
| SCHOOL TYPE | -14.276 | 2.113 | <0.001*** | -23.325 | 2.330 | <0.001*** |
| REHEARSAL | -2.304 | 2.893 | 0.426 | -5.589 | 3.649 | 0.126 |
| ELABORATION | -1.826 | 5.146 | 0.723 | 3.603 | 6.381 | 0.573 |
| ORGANIZATION | 2.291 | 4.536 | 0.614 | -2.471 | 5.733 | 0.667 |
| METACOGNITION | 11.435 | 4.685 | 0.015* | 16.072 | 5.602 | 0.004** |
| URBAN | 1.862 | 2.125 | 0.382 | 6.997 | 2.198 | 0.002** |
| SUBURBAN | 1.576 | 2.073 | 0.448 | 0.116 | 2.259 | 0.959 |
| For GENDER slope, $\beta_1$ | | | | | | |
| INTRCPT2 | -10.965 | 2.960 | <0.001*** | -4.083 | 0.953 | <0.001*** |
| SCHOOL TYPE | 3.899 | 2.616 | 0.137 | | | |
| URBAN | 0.738 | 2.239 | 0.742 | | | |
| SUBURBAN | -3.479 | 2.464 | 0.159 | | | |
| For REHEARSAL slope, $\beta_2$ | | | | | | |
| INTRCPT2 | -1.126 | 0.587 | 0.055 | -2.035 | 0.676 | 0.003** |
| For ELABORATION slope, $\beta_3$ | | | | | | |
| INTRCPT2 | 5.102 | 0.964 | <0.001*** | 6.613 | 1.053 | <0.001*** |
| For ORGANIZATION slope, $\beta_4$ | | | | | | |
| INTRCPT2 | -5.500 | 2.778 | 0.048* | -1.602 | 1.052 | 0.128 |
| SCHOOL TYPE | 3.597 | 2.631 | 0.172 | | | |
| URBAN | -1.563 | 1.580 | 0.323 | | | |
| SUBURBAN | -0.093 | 1.736 | 0.957 | | | |
| For METACOGNITION slope, $\beta_5$ | | | | | | |
| INTRCPT2 | 5.677 | 0.976 | <0.001*** | 1.000 | 2.778 | 0.719 |
| SCHOOL TYPE | | | | 3.946 | 2.656 | 0.138 |
| URBAN | | | | 2.690 | 1.687 | 0.112 |
| SUBURBAN | | | | 0.955 | 1.865 | 0.609 |

*Note*. Final estimates were calculated based on the robust standard error.

* $p < 0.05$,

** $p < 0.01$, and

***$p < 0.001$.

We found that students' *literacy* achievement was lower on average in public schools than in private schools ($\gamma$ = -14.276, $p$ < .001), holding constant other school-level and student-level predictors. Moreover, *literacy* achievement of *female* students was significantly higher than that of *male* students on average ($\gamma$ = -10.965, $p$ < .001), holding constant other school-level and student-level predictors. The directions and tests of between- and within-school effects remained the same as those before adding the school-level predictors.

**CS on math model.** With the same model structure, we investigated how cognitive learning strategies predicted students' *math* achievement with the initial HLM analysis. *Metacognition* positively predicted students' *math* achievement both within and between schools. This suggests that for students on average across schools, when their *metacognition* increased, *math* achievement tended to increase, holding constant other predictors ($\gamma$ = 6.323, $p$ < .001); also, schools with higher average levels of *metacognition* tended to show higher average levels of *math* achievement ($\gamma$ = 20.234, $p$ < .01). *Rehearsal* and *elaboration* significantly explained within-school difference, however, not between-school difference in *math* achievement, holding constant other predictors. On average across schools, when students' *elaboration* increased, their *math* achievement also tended to increase ($\gamma$ = 6.604, $p$ < .001); when students' *rehearsal* increased, however, their *math* achievement tended to decrease ($\gamma$ = -2.014, $p$ < .01), controlling other predictors. *Organization* did not show significant between-school effect or average within-school effect on math achievement. Also, *math* achievement of *male* students was significantly lower than *female* students on average ($\gamma$ = -4.083, $p$ < .001), holding constant other school-level and student-level predictors.

The effects of *metacognition* significantly varied across schools. Thus, we added school-level predictors in the final *CS on Math Model* (Table 3). The results showed that students' *math* achievement was lower on average in *public* schools than in *private* schools ($\gamma$ = -23.325, $p$ < .001), and *urban* schools had higher levels of *math* achievement than *rural* schools ($\gamma$ = 7.00, $p$ < .01), holding constant other school-level and student-level predictors. Interestingly, *metacognition* no longer had a significant effect on *math* achievement controlling for school type and school location. The directions and tests of between- and within-school effects of other cognitive learning strategy predictors remained the same as those before adding the school-level predictors.

**BS on literacy model.** Among different behavioral learning strategies, only *effort-regulation* positively predicted students' *literacy* achievement both within and between schools in the initial HLM model. This suggests that on average across schools, as *effort-regulation* increased, students' *literacy* achievement tended to increase, holding constant other predictors ($\gamma$ = 10.934, $p$ < .001); also, schools with higher average levels of *effort-regulation* were related to higher average levels of literacy achievement ($\gamma$ = 16.076, $p$ < .001). *Time-regulation* and *environment-regulation* significantly explained within-school difference yet not between-school difference in *literacy* achievement, holding constant other predictors. More specifically, on average across schools, when students' *time-regulation* increased, their *literacy* achievement tended to decrease ($\gamma$ = -7.85, $p$ < .001); in contrast, as students' *environment-regulation* increased, *literacy* achievement tended to increase ($\gamma$ = 4.514, $p$ < .001), holding constant other predictors. *Help-seeking* and *task-regulation* had significant between-school effects. Schools with lower levels of *help-seeking* and higher levels of *task-regulation* tended to have higher levels of *literacy* achievement, holding constant other predictors. Moreover, *literacy* achievement of *males* was significantly lower than that of *females* on average ($\gamma$ = -7.308, $p$ < .001), holding constant other school-level and student-level predictors.

The effects of *environment-regulation* and *help-seeking* significantly varied across schools. Thus, we added school-level predictors in the final *BS on Literacy Model* (Table 4). We found that students' *literacy* achievement was lower on average in *public* schools than in *private*

**Table 4. HLM results of behavioral strategy use and academic achievement.**

| Fixed Effect | Behavioral Strategy on Literacy Final Model | | | Behavioral Strategy on Math Final Model | | |
|---|---|---|---|---|---|---|
| | $B$ | $SDs$ | $p$ | $B$ | $SDs$ | $p$ |
| For INTRCPT1, $\beta_0$ | | | | | | |
| INTRCPT2 | 175.262 | 13.574 | <0.001*** | 161.775 | 14.682 | <0.001*** |
| SCHOOL TYPE | -12.521 | 2.346 | <0.001*** | -23.509 | 2.414 | <0.001*** |
| TASK REGULATION | 10.663 | 4.652 | 0.022* | 13.240 | 6.639 | 0.047* |
| EFFORT REGULATION | 15.266 | 4.925 | 0.002** | 16.621 | 6.366 | 0.009** |
| TIME REGULATION | -7.711 | 4.450 | 0.084 | -5.525 | 5.297 | 0.352 |
| ENVIRONMENT REGULATION | 4.176 | 4.338 | 0.336 | 7.721 | 5.378 | 0.152 |
| HELP SEEKING | -9.946 | 5.073 | 0.051 | -12.328 | 5.787 | 0.034* |
| URBAN | 1.344 | 2.016 | 0.505 | 5.409 | 2.180 | 0.013* |
| SUBURBAN | -0.627 | 1.930 | 0.746 | -0.789 | 2.205 | 0.721 |
| For GENDER slope, $\beta_1$ | | | | | | |
| INTRCPT2 | -7.308 | 0.824 | <0.001*** | -2.953 | 0.902 | 0.001** |
| For TASK REGULATION slope, $\beta_2$ | | | | | | |
| INTRCPT2 | 0.4383 | 0.9440 | 0.642 | 0.217 | 1.037 | 0.834 |
| For EFFORT REGULATION slope, $\beta_3$ | | | | | | |
| INTRCPT2 | 10.919 | 1.077 | <0.001*** | 4.123 | 3.387 | 0.283 |
| SCHOOL TYPE | | | | 5.428 | 3.652 | 0.138 |
| URBAN | | | | 4.037 | 2.180 | 0.065 |
| SUBURBAN | | | | 3.209 | 2.415 | 0.185 |
| For TIME REGULATION slope, $\beta_4$ | | | | | | |
| INTRCPT2 | -7.870 | 0.965 | <0.001*** | -2.288 | 1.092 | 0.036* |
| For ENVIRONMENTAL REGULATION slope, $\beta_5$ | | | | | | |
| INTRCPT2 | 2.477 | 3.643 | 0.497 | 5.360 | 0.938 | <0.001*** |
| SCHOOL TYPE | 1.851 | 3.191 | 0.562 | | | |
| URBAN | -0.384 | 2.094 | 0.854 | | | |
| SUBURBAN | 1.121 | 2.343 | 0.633 | | | |
| For HELP_SEEKING slope, $\beta_6$ | | | | | | |
| INTRCPT2 | -3.538 | 5.081 | 0.487 | -1.348 | 1.167 | 0.248 |
| SCHOOL TYPE | 0.726 | 4.691 | 0.877 | | | |
| URBAN | 5.797 | 2.411 | 0.017* | | | |
| SUBURBAN | 5.077 | 2.427 | 0.037* | | | |

*Note*. Final estimates were calculated based on the robust standard error.

* $p < 0.05$,

** $p < 0.01$, and

***$p < 0.001$.

schools ($\gamma = -12.520$, $p < .001$), holding constant other school-level and student-level predictors. Furthermore, within-school effects of *help-seeking* varied by school locations. Specifically, students in *urban* and *suburban* schools had positive average effects whereas those in *rural* schools had a negative average effect on *literacy* achievement, controlling for school type and other student-level predictors.

**CS on math model.** In the initial HLM model of how students' behavioral learning strategies predicted their *math* achievement, e*ffort-regulation* predicted students' *math* achievement both within and between schools. This means that on average across schools, when students had higher *effort-regulation*, their *math* achievement tended to increase, holding constant

other predictors ($\gamma$ = 12.194, $p < .001$); also, schools with higher average levels of *effort-regulation* tended to show higher average levels of *math* achievement ($\gamma$ = 18.384, $p < .01$). *Time-regulation* and *environment-regulation* also significantly explained within-school difference yet not between-school difference in *math* achievement, holding constant other predictors. On average across schools, when students' *environment-regulation* increased, their *math* achievement also tended to increase ($\gamma$ = 5.382, $p < .001$), but when students' *time-regulation* increased, their *math* achievement tended to decrease ($\gamma$ = -2.266, $p < .05$), while controlling for other predictors. *Help-seeking* did not have a significant within-school effect, but it had a significant between-school effect ($\gamma$ = -12.330, $p < .05$) on *math* achievement. Also, *math* achievement of *male* students was significantly lower than *female* students on average ($\gamma$ = -2.953, $p < .01$), holding constant other school-level and student-level predictors.

The within-school effect of *effort-regulation* significantly varied across schools. Thus, we added school-level predictors in the final *BS on Math Model* (Table 4). The results showed that students' *math* achievement was lower on average in *public* schools than in *private* schools ($\gamma$ = -23.509, $p < .001$), holding constant other school-level and student-level predictors. In addition, *urban* schools had higher *math* achievement than *rural* schools controlling for other predictors ($\gamma$ = 5.409, $p < .05$). The directions and tests of between- and within-school effects remained the same as those before adding the school-level predictors.

## Discussion

This study showed that successful self-regulated adult and elementary learners might differ in their SRL strategy use due to developmental issues. Within the school level, certain SRL strategies used by elementary learners (CS: metacognition; BS: task-regulation and effort regulation) were linked to higher achievement scores; however, other strategies (CS: rehearsal, elaboration, and organization; BS: time regulation and environment regulation) were not significant predictors in this study. This suggests that learning experiences in schools need to support elementary learners in developing their SRL strategy use. Furthermore, we found that the relationship between students' SRL strategy use and their academic achievement is more strongly explicable at the individual level. Nonetheless, the variances of schools that can still be attributed to 14–19 percent. This indicates that both the individual and the school environmental learning support should be taken into account for elementary students' SRL growth.

First, regarding RQ 1, we explored the relationship between elementary students' CS use and academic achievement. We found that South Korean elementary students' metacognitive strategy use is an important learning skill supporting academic performance in literacy and math; it was a significant predictor of academic achievement both within and across the schools. Many scholars also have highlighted the important role of metacognition in controlling effective cognitive processing through planning, monitoring, and evaluation in the adolescent and adult years [5, 29, 48, 49]. Students who review their thinking processes in subject learning may solve problems more effectively. In an average school, some relationship patterns were different from the within-school level. For example, students' elaboration and organization use were related to higher literacy achievement; students' elaboration use was positively linked to math achievement; however, rehearsal was negatively linked to math achievement. This complex pattern has also been reported in a meta-study of adult learners by Broadbent and Poon [49]; in this study, university students' metacognition predicts higher achievement; however, elaboration and rehearsal were not. Related to the different relationships between the cognitive strategies use and learning outcomes, Weinstein, Acee [31] pointed out that using rehearsal as a lower-level cognitive strategy may differ from higher-level strategies regarding achievement.

Next, regarding RQ 2, we explored the relationship between elementary students' BS use and academic achievement. We found that students with higher academic achievement tended to regulate their effort depending on task difficulty. Kim, Park [48] reported similar results at the high school level, such that students' effort-regulation was related to their academic achievement, with this linear relationship stronger in a group of high achievers. However, within a school level, we found time-regulation and environment-regulation strategies use did not predict better achievement. Also, interestingly, this study's results showed that when students frequently use the help-seeking strategy, their math scores tend to be lower than those who do not seek additional help. Related to the cultural background, Loh and Teo [50] pointed out that in Eastern culture, students tend to be more passive learners and feel difficulty asking questions to others when they need help. Age differences may cause this result; for example, Broadbent and Poon (49) reported that time-regulation and effort-regulation use were related to the higher achievement of university students in online learning contexts; on the other hand, help-seeking was not. On average across schools, those relationship patterns between BS use and academic achievement differed from the results within schools. Elementary students' time-regulation was both negatively related to literacy and math achievement. Young students may not have developed sufficient skills in time management for their difficult learning tasks. Across schools, help-seeking strategy was not a significant predictor of both literacy and math achievement. The variation in students' help-seeking strategy use at the individual level can be attributed to differences in their communication styles and personalities.

In addition, regarding RQ 3 and 4, we explored school environmental factors on students' SRL strategy use and academic achievement. In this study, the average literacy and math scores were both high in private schools. This is not surprising because, in South Korea, a high proportion of students with low achievement belong to low-income families and students in these families mostly enter public elementary schools [51]. Many previous studies pointed out that social economic status of family related to students' academic achievement [52–54]. We also found that urban school students in South Korea had higher math achievement when controlling all other learning strategies and gender differences; however, there were no differences related to school locations in literacy achievement. Interestingly, rural students tended to use less help-seeking strategies than others, which might relate to students' perceptions of insufficient external resources. The result highlights that further research is needed to explore the educational inequity issue for underprivileged students (e.g., in low-income families or rural areas).

Lastly, regarding the gender variable, this study's results showed interesting findings. Females of 6th-graders performed better than males in literacy and math. Gender differences in academic achievement may be caused by different growth patterns in children's linguistic and cognitive development [55]. In general, females at the high school and undergraduate levels show a lower preference for STEM (i.e., Science, Technology, Engineering, and Mathematics) majors and higher anxiety about learning in it [56, 57]. Our study supports previous research suggesting that positive teacher and parent expectations and additional learning opportunities during secondary school may help females develop a sense of confidence and expectation for success in STEM fields in the future.

## Limitations and future research

As a cross-sectional study, this study may include some limitations. The study's results partly demonstrated that the use of self-regulated learning strategies by 6th-grade students is not entirely identical to the pattern seen in adult learners. Considering previous findings with university-level students [38–40], the relationship patterns between young students' SRL strategy

use on academic achievement were different. However, to provide a better insight into learning development, the follow-up studies need to track the patterns of SRL strategy use on academic achievement in middle and high school through the longitudinal study design.

When we conducted preliminary analyses, the four cognitive and five behavioral strategies were highly correlated with each other. To avoid multicollinearity issues and build a parsimonious model avoiding high-complexity, we decided to build two separate HLM models in cognitive and behavioral strategies. However, future research could explore the complex relationships between the two strategies. The different statistic models under the combined concept of SRL strategies would provide insightful suggestions for elementary education.

In this study, we estimated school environmental factors (i.e., school types and locations) to interpret the results in South Korean educational contexts. The national database selected samples by strategic sampling considering the general Korean student population. Because most South Korean elementary schools belong to the public school system, the population between public and private schools was not balanced in this study. However, we followed the suggestion that unequal group size is not a problem in both groups having large samples [45] and have more benefits related to strategic sampling considering the population. This context should be fully considered in the interpretation of the results.

## Conclusion

Our results raise important implications for supporting SRL strategies in K-12 education. Elementary students need supportive learning opportunities to expand their use of SRL strategies and grow into successful self-regulated adult learners. Teachers need to help children learn to plan, monitor, and evaluate their learning processes during classroom activities. Promoting SRL strategies can help elementary students increase their learning efficiency. Also, understanding those different SRL developmental needs according to school environments can contribute to equity in education by developing important research questions for underprivileged students.

## Supporting information

**S1 Appendix. HLM equation.**
(DOCX)

## Author Contributions

**Conceptualization:** Cheyeon Ha, Alysia D. Roehrig.

**Data curation:** Cheyeon Ha.

**Formal analysis:** Cheyeon Ha, Qian Zhang.

**Methodology:** Cheyeon Ha, Qian Zhang.

**Resources:** Cheyeon Ha.

**Software:** Cheyeon Ha, Qian Zhang.

**Supervision:** Alysia D. Roehrig.

**Writing – original draft:** Cheyeon Ha, Alysia D. Roehrig, Qian Zhang.

**Writing – review & editing:** Cheyeon Ha, Alysia D. Roehrig, Qian Zhang.

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
