## [Decision Letter · Decision Letter 0]

31 Jan 2023

PONE-D-22-33255Self-regulated Learning Skills and Academic Achievement in South Korean 6th-graders: Based on Varying School EnvironmentsPLOS ONE

Dear Dr. Ha,

Thank you for submitting your manuscript to PLOS ONE. After careful consideration, we feel that it has merit but does not fully meet PLOS ONE’s publication criteria as it currently stands. Therefore, we invite you to submit a revised version of the manuscript that addresses the points raised during the review process. Please submit your revised manuscript by Mar 17 2023 11:59PM. If you will need more time than this to complete your revisions, please reply to this message or contact the journal office at plosone@plos.org. Please include the following items when submitting your revised manuscript:A rebuttal letter that responds to each point raised by the academic editor and reviewer(s). You should upload this letter as a separate file labeled 'Response to Reviewers'.A marked-up copy of your manuscript that highlights changes made to the original version. You should upload this as a separate file labeled 'Revised Manuscript with Track Changes'.An unmarked version of your revised paper without tracked changes. You should upload this as a separate file labeled 'Manuscript'.

We look forward to receiving your revised manuscript.

Kind regards,

Arslan Ayub

Academic Editor

PLOS ONE

Journal Requirements:

Additional Editor Comments:

Dear Authors,  You are required to address reservations raised by each reviewers. Be specific in your response and indicate changes in the manuscript.  Good luck! 

Reviewers' comments:

Reviewer's Responses to Questions

**Comments to the Author**

1. Is the manuscript technically sound, and do the data support the conclusions?

Reviewer #1: Yes

Reviewer #2: No

Reviewer #3: Partly

Reviewer #4: Yes

2. Has the statistical analysis been performed appropriately and rigorously? 

Reviewer #1: Yes

Reviewer #2: Yes

Reviewer #3: Yes

Reviewer #4: Yes

3. Have the authors made all data underlying the findings in their manuscript fully available?

Reviewer #1: Yes

Reviewer #2: Yes

Reviewer #3: Yes

Reviewer #4: Yes

4. Is the manuscript presented in an intelligible fashion and written in standard English?

Reviewer #1: Yes

Reviewer #2: No

Reviewer #3: Yes

Reviewer #4: Yes

5. Review Comments to the Author

Reviewer #1: This is a significant study on a large data base (longitudinal). While the study has a lot of merits, I would like to invite the authors to clarify the following issues through revision:

1. What are the strengths of exploring only ‘the cognitive and behavioral learning strategies from Pintrich’s SRL model (not with motivation)?

2. What is the KELS database about? I think that the database needs to be described when it was mentioned for the first time in the main text.

3. What do the authors mean by individual differences in self-regulation？

4. The writing needs to present a much more compelling argument for the research questions to be addressed in the inquiry. It looks that these questions have already been examined and answered in previous research.

5. Please indicate how each research question was addressed by the results in the findings’ section.

6. ‘Twenty-first century learners need to have learning opportunities to activate various…’. Does this mean that learners in other centuries do not need?

7. I think that the findings should add to what we know about self regulation and academic achievements. Not sure if the study adds to our understanding of self regulation and academic achievements. The discussion needs some explicit comparison between the study’s findings and those of previous research. Apart from confirming those of previous research, what else did the study achieve?

Reviewer #2: This is a cross-sectional study. For the current report, authors used the Korean ELS Study and report if the association between some cognitive elements and kids scores vary based on the context they are present. The context on this study is mainly private vs public school. Although the sample size is large, and there were opportunities for a great paper, authors did not take the best advantage of such a great data. Here are my main concerns about this paper:

1- The main reason I would be interested in this paper would be the groups differences not the main effects. This includes main effects of school. Of course, we know that higher cognitive capacity is good. Of course, we know that private schools are more successful than public schools. These are questions that have been answered decades ago. What would be relevant and interesting here are not those questions but whether these processes vary based on the context. Unfortunately, this is only superficially touched, and there is no good conversation / discussion on this topic. For example, the conclusion is full of general points irrelevant to school context.

2- Regarding the structure, we need to see limitations and future directions (not the reasons of conflicting results with the literature) in such sections. We need to see the aims of the study at the end of the background / intro, not the 2nd paragraph of the intro. We need to see more specific terms in the conclusion. We wish to see the summary of the study findings in the beginning of the discussion.

3- Regarding the measures, reliability is given, but the range of the score, the thresholds used, and the meaning of the scores are not given. There are no indications of the validity either.

4- Regarding the tables, in any section that presents data by group, we need to see p values. We also need to see the name o the statistical test below that table. Currently, only regression tables give p values.

Reviewer #3: Abstract

The abstract provides a clear overview of the study’s aim to explain the relationships between self-regulated learning strategy use and academic achievement of 6th-grade students in South Korea. It also mentions the use of a large existing database, the Korean Educational Longitudinal Study (KELS), which allows for a detailed analysis of the relationship between self-regulated learning strategy use and academic achievement. The use of a large dataset is a strength as it allows for a more representative sample and increases the generalizability of the findings.

The abstract also mentions the use of 2-level hierarchical linear models (HLM), which is an appropriate statistical method for analyzing data with a nested structure, such as students within schools. Additionally, the abstract highlights the consideration of environmental factors such as school type and location, which is a valuable aspect of the study as it allows for a more comprehensive understanding of the relationship between self-regulated learning strategy use and academic achievement.

The abstract also presents the study’s key findings, including that metacognition and effort regulation positively predicted literacy and math achievement and that average literacy and math achievement were significantly higher in private schools than in public schools. However, the abstract could be improved by briefly explaining the study’s limitations and novelty. This is important as there are several studies on math and self-regulation. Overall, the abstract is clear and summarises the study’s aims, methods, and key findings well.

Introduction

The author presents a well-written and detailed background on the importance of self-regulation skills in learning and the lack of emphasis on teaching these skills in the K-12 curriculum. They also provide a clear overview of the conceptual frameworks and previous research on self-regulated learning (SRL) strategies in elementary students, highlighting the limitations of small sample sizes in past studies. The use of the Korean Longitudinal Educational Study (KELS) database, which offers a large and nationally representative sample, is a strength of the study. The author also thoroughly reviews the relevant literature and previous studies using the KELS database. However, it would be beneficial for the author to provide more specific details on the research question and methodology of the current study and how it will contribute to filling the gap in the literature on SRL in elementary students. The novelty should be stated in the introduction specifically, as it was not stated in the abstract.

Literature review

The author provides a thorough and well-written literature review on the learning strategies of self-regulated learners. They clearly explain the importance of self-regulation in academic achievement and cite relevant theories and studies in the field. The use of Pintrich’s model of self-regulated learning and the organization of the strategies into cognitive and behavioural categories is a strength of the review. The author also explains the specific types of cognitive strategies, such as rehearsal and elaboration, and behavioural strategies, such as time and effort regulation, and how they relate to self-regulation and academic achievement. However, the literature review could be improved by including more recent studies and providing a more detailed synthesis of the literature to better identify any gaps or inconsistencies in the literature and how the current study aims to address them.

The research problem presents a novel contribution to the literature by investigating the relationship between South Korean 6th-graders’ use of self-regulated learning (SRL) strategies and their academic achievement in literacy and math. The study is also unique in that it specifically focuses on the early developmental stage of SRL strategies in students of this age group. The use of Pintrich’s model of SRL and the inclusion of both cognitive and behavioural strategies as the features of successful self-regulation is also noteworthy. The research questions, specifically Q1 and Q2, aimed to investigate the relationship between SRL strategy use and academic achievement, while Q3 and Q4 aimed to investigate the moderating effect of school type and location on this relationship, which provides some novelty. Overall, this research problem presents a novel and valuable addition to the field as it aims to fill the gap in the literature on SRL strategies in early developmental stage students in a specific cultural context.

Method

The method presented is comprehensive and includes detailed information on the data sources, study design, and measures used in the study. The use of a national dataset, the Korean Longitudinal Educational Study (KELS), is a strength of the study as it allows for a large and nationally representative sample. The inclusion of both cognitive and behavioural strategies, as well as the use of the MSLQ developed by Pintrich, provides a solid foundation for the study. The use of 2-level HLM analyses to examine predictors and moderators at the student and school levels is also appropriate for this study. The exclusion of missing data and the final sample size is also reported. The inclusion of demographic information such as gender and school location and the use of a vertical scaling approach to adjust the academic achievement scores are also noteworthy. However, it would be beneficial for the author to provide more information on the specific statistical tests used and the criteria for missing data exclusion. Additionally, it would be beneficial to provide more information on how the sample was selected, if it was a random sample or not and if the sample is representative of the population. In addition, and more importantly, the original scale for Pintrich was a 7-point scale. Please justify the 4-point scale used.

Results

The reported results are well presented.

Discussion

The discussion in this journal article presents a thorough analysis of the study’s results and their implications. The authors make a strong argument for the importance of SRL strategies in academic performance, particularly for young learners. They also critically examine the relationship between different types of SRL strategies and academic achievement, explaining why some strategies may have a stronger impact on performance than others. Additionally, they discuss the potential moderating effects of school location and type on this relationship.

The authors use previous studies on the topic to support their arguments and provide context for their findings. They also highlight the limitations of the study and potential areas for future research. Overall, the discussion demonstrates a sound understanding of the theoretical foundations of the study and a critical evaluation of the results.

Conclusion

The conclusion is clear and flows well. It highlights the importance of SRL strategy use as a key learning skill for future generations, and the study’s findings have important implications for future research and practice in K-12 education. The conclusion suggests that elementary students should have more opportunities to develop SRL skills and that schools should provide an appropriate learning experience to help students develop these skills well. The conclusion also highlights the importance of metacognitive strategies in learning and the need for teachers to help children learn to plan, monitor, and evaluate their learning processes during classroom activities. The conclusion also notes the need for educational researchers to work closely with teachers to incorporate these theoretical principles into specific subject curricula.

The conclusion also points out that teachers’ roles in guiding SRL strategies need to be better supported in school environments, including public schools and rural areas, where achievement gaps are evident. It also highlights that school location was an influential factor in mathematical achievement, suggesting that there is a particular need to support SRL strategies for rural students in math education to prepare them for future STEM education. The conclusion also suggests that promoting metacognition, task-regulation, and effort-regulation learning strategies could help young students to increase their learning efficiency. Overall, the conclusion makes a valuable contribution to the knowledge of the field by providing practical recommendations based on the study’s findings.

The study also found a notable achievement gap between public and private schools in South Korea and highlights that future research should consider students’ socioeconomic background to explain the gaps in achievement and learning strategies. This research can serve an important social justice purpose by addressing equity issues, and as such, it should be highlighted.

There are a few typo errors noted in the track changes manuscript. Please address.

Reviewer #4: The manuscript titled “Self-regulated Learning Skills and Academic Achievement in South Korean 6th-graders Based on Varying School Environments” is an interesting study, that complies with the PLOS ONE standards. It´s an interesting, well conducted work regarding the Learning Skills and Academic Achievement area of emphasis. However, it has some weaknesses that should be considered.

Overall, my main concern is:

I consider the document too long, making it difficult to read and understand its content. I believe that it would be beneficial to synthesize some parts of the article (I will mention them in the specific considerations), which would make the document more concise and, consequently, easier to read and understand.

In addition, the authors must correct the entire document, regarding the bibliographic references. They used the “APA” norms, when the formatting of the journal requires “Vancouver”.

Some specific considerations:

Introduction / Background:

Although the content of this chapter is adequate, I propose to summarize. There is too much information, which is not used later, namely in the discussion of the results. Basically, the theoretical framework has an excess of information and content (although adequate). I propose that the authors limit themselves to the essential content, which should be restricted to what is used for the discussion of the results.

On page 9 (line 13), there is a reference without date.

Study Purpose and Research Questions:

The first sentence of this chapter needs to be changed. It's very confusing.

Method:

In the “Data Analysis” part, an exhaustive description of the statistical methods used in all the data analysis that was carried out is required. They must describe all the statistical techniques used as well as the significance levels.

Results:

The content of this chapter should also be more abbreviated. Namely, authors should limit themselves to presenting the results and, at most, highlight some results. What happens is that, at times, the results are practically discussed, which should be referred to the discussion chapter.

I do not agree that the comparison between public and private schools (School Types) can exist and be used in this study. This observation has to do with the fact that, out of a total of 446 schools, only 12 are public (2.7% of the sample). I propose that this comparison be withdrawn.

Discussion:

At the beginning of this chapter, please recall the main objective of the study and also the main findings.

Conclusion:

Once again, the conclusion has too much information. I suggest that the authors limit themselves to drawing the conclusions arising from the analyzes carried out on their results and ignore everything else

6. PLOS authors have the option to publish the peer review history of their article (what does this mean?). If published, this will include your full peer review and any attached files.

Reviewer #1: No

Reviewer #2: No

Reviewer #3: No

Reviewer #4: No

---

## [Author Response · Author response to Decision Letter 0]

23 Mar 2023

We uploaded the word document (Response to Reviewers) on the submission system which included detailed revision comments.

---

## [Editor Report · Decision Letter 1]

29 Mar 2023

Self-regulated Learning Strategies and Academic Achievement in South Korean 6th-graders: A Two-level Hierarchical Linear Modeling Analysis

PONE-D-22-33255R1

Dear Dr. Authors,

We’re pleased to inform you that your manuscript has been judged scientifically suitable for publication and will be formally accepted for publication once it meets all outstanding technical requirements.

Kind regards,

Arslan Ayub

Academic Editor

PLOS ONE
---

## [Editor Report · Acceptance letter]

13 Apr 2023

PONE-D-22-33255R1 

Self-regulated Learning Strategies and Academic Achievement in South Korean 6th-graders: A Two-level Hierarchical Linear Modeling Analysis 

Dear Dr. Ha:

I'm pleased to inform you that your manuscript has been deemed suitable for publication in PLOS ONE. Congratulations! Your manuscript is now with our production department. 

Kind regards, 

on behalf of

Dr. Arslan Ayub 

Academic Editor

PLOS ONE